# Geographical and social isolation drive the evolution of Austronesian languages

**Cecilia Padilla-Iglesias**[1]*, **Erik Gjesfjeld**[2], **Lucio Vinicius**[1]*

**1** Department of Anthropology, University of Zurich, Zurich, Switzerland, **2** Department of Archaeology, University of Cambridge, Cambridge, United Kingdom

* cecilia.padillaiglesias@uzh.ch (CPL); Lucio.vinicius@aim.uzh.ch (LV)

**Data Availability Statement:** All relevant data are within the paper and its Supporting Information files.

**Funding:** The author(s) received no specific funding for this work.

## Abstract

The origins of linguistic diversity remain controversial. Studies disagree on whether group features such as population size or social structure accelerate or decelerate linguistic differentiation. While some analyses of between-group factors highlight the role of geographical isolation and reduced linguistic exchange in differentiation, others suggest that linguistic divergence is driven primarily by warfare among neighbouring groups and the use of language as marker of group identity. Here we provide the first integrated test of the effects of five historical sociodemographic and geographic variables on three measures of linguistic diversification among 50 Austronesian languages: rates of word gain, loss and overall lexical turnover. We control for their shared evolutionary histories through a time-calibrated phylogenetic sister-pairs approach. Results show that languages spoken in larger communities create new words at a faster pace. Within-group conflict promotes linguistic differentiation by increasing word loss, while warfare hinders linguistic differentiation by decreasing both rates of word gain and loss. Finally, we show that geographical isolation is a strong driver of lexical evolution mainly due to a considerable drift-driven acceleration in rates of word loss. We conclude that the motor of extreme linguistic diversity in Austronesia may have been the dispersal of populations across relatively isolated islands, favouring strong cultural ties amongst societies instead of warfare and cultural group marking.

## Introduction

Languages are the product of long-term cumulative cultural evolution [1]. As the product of social learning, transmission and use, languages respond to selection pressures posed by local communication contexts deriving from physical, social and cognitive environments [2,3]. Although evolutionary changes have resulted in a vast array of over 7000 languages spoken in the world today [4], the questions of "why so many languages" and "why so unevenly distributed" are far from being answered. Nonetheless, a vast corpus of research has emerged on the evolutionary, ecological and social correlates of the global distribution of linguistic diversity [5–15].

Various studies have focused on demography as a key determinant of rates of cultural and linguistic evolution [16–22]. The most discussed demographic factor is population size, but

**Competing interests:** The authors have declared that no competing interests exist.

studies disagree on its effects. Some analyses suggest that larger populations promote innovation, are less prone to cultural drift and random loss of linguistic elements, and may exhibit less stringent enforcement of norms, thus allowing languages to change faster [1,16,17,19,23–25]. In contrast, other studies have argued that linguistic differentiation should be fastest in small populations due more rapid diffusion of new features [10], greater tolerance of diversity [26], and stronger response to contact resulting from trade and marriage across groups [24]. Recent models have added population structure as an essential demographic factor underlying cultural and linguistic evolution. For example, population density, local interconnectedness and migrations were claimed to play a key role in cumulative cultural evolution by facilitating the emergence, diffusion and survival of linguistic innovations [18,21,27–30]. In contrast, social, cultural and political settings [31,32] might also promote social group cohesion and the sharing of linguistic features within groups, which would reduce rates of linguistic differentiation [33].

In addition to social and demographic factors, geographical isolation may also constrain communication between populations and reduce diffusion of linguistic or cultural traits, increasing the likelihood of random losses of cultural items [17,34], in a process analogous to genetic drift [35]. However, lost items may be replaced by novel innovations that would be more easily fixated [36,37]. In contrast, other studies proposed that contact is more effective than isolation in promoting language differentiation, as it facilitates the introduction of novel linguistic forms via second language acquisition [38].

Another factor often associated with linguistic diversification is warfare. From this perspective, conflict between groups may foster deliberate linguistic differentiation [39,40] as the result of cultural group marking [41,42]. In other words, whilst the use of language as an index of cultural group identity is widespread [39,40,43,44], experimental paradigms [42], computational simulations [41] and ethnographic accounts [45] suggest that this indexical function of language may be particularly salient when neighbouring groups are in conflict with one another, as the ability to distinguish in-groups from out-groups may become a matter of life or death. Thomason [29] describes several examples of this last phenomenon, such as the meeting that a European missionary attended during the 17th century at which the Delaware Indians planned to substitute different words for their native lexicon when they went to war against the Iroquois, so that their enemies wouldn't understand them. A similar trend was observed among sixteenth-century Portuguese speakers trying to deliberately differentiate their language from that of their Spanish opponents. A related argument for a prominent role of between-group conflict is cultural group selection [46,47], which proposes that linguistic diversity reflects the relative success of competing cultural groups and thus ascribes a crucial role to warfare, extinction and acculturation in the distribution of languages [47]. However, empirical validation of those claims remains limited.

One of the reasons for the ongoing debate over language diversification is that previous studies have not yet attempted to include all the key geographic, social and demographic variables discussed above in a single analysis. This task is even more challenging since many of these factors act simultaneously and tend to be more similar between closely related languages [22,48]. In addition, most have used contemporary speaker population sizes in order to explain linguistic evolution over many years [22,48]. This approach is in general problematic for establishing a direction of causality, and in particular in Austronesia given that the demographic, socioeconomic, and cultural landscape has dramatically changed over the past century first due to European colonialism and imposed linguistic policies[49], and then due to the current globalising trends [49–51] as well as high rates of population growth [48]. Hence, if we wish to evaluate the factors that have affected the evolution of Austronesian languages, they should ideally reflect those societies prior to colonial times (see Bromham et al. [48]). Here we provide the first integrated test of the effect of various socio-demographic and geographic features on

linguistic diversification among 50 Austronesian languages. The Austronesian family is the second most diverse linguistic group in the world with nearly 1200 languages and has been extensively investigated [40,52]. Austronesian languages tend to be restricted to clearly defined islands or archipelagos across the Pacific and Indian oceans [49], which offers an ideal context to investigate how geography affects genetic or cultural diversity [53].

We obtained information concerning the historical (prior to large-scale modernisation) state of five variables: population size, geographical isolation, within-group conflict, between-group conflict (same cultural group), and between-group conflict (distinct cultural groups; Table 1) from 50 Austronesian ethnolinguistic groups (Fig 1). They were used in Poisson generalized linear models as predictors of three measures of linguistic diversification: The rate at which new basic vocabulary items are added to a languages' vocabulary (word gains), the rate at which existing basic vocabulary items are lost (word losses), and the overall effect of these two processes in the divergence of vocabularies between pairs of sister languages (lexical turnover). Either a higher rate of word gains or word losses result in sister languages having less words in common to define a pre-specified list of concepts (see Methods and Greenhill et al. [22] for a use of the same method). We applied Bayesian Poisson regressions to our 25 sister pairs using a time-calibrated phylogenetic sister-pairs approach [22,54] to control for shared evolutionary histories and overcome the problem of statistical non-independence that often characterises comparative studies of cultural and linguistic evolution [47].

Different from other approaches to evolutionary rates (such as least-squares regressions based on data diagnosed by Welch and Waxman tests), the sister-pairs approach does not increase uncertainty for more closely related languages, excludes data points with relatively

**Table 1. Predictor variables used in our analyses.** All predictors were taken from the Pulotu dataset [82]. In addition to the specified transformations, all predictors were standardised (we subtracted their means and divided the result by their standard deviation).

| Original Variable Name | Original Variable Description | Original Scale | Transformation |
|---|---|---|---|
| Distance to closest landmass inhabited by a different culture | A different culture is any culture other than the culture being coded. The distance stated is a minimum distance. If there was a different culture living on the same island, code this distance as "0". | kilometres | Log-transformed |
| (No) conflict (social or political) within the local community | Conflict within the local community can include both interpersonal and intergroup conflict. Only conflict that poses a realistic threat to the cohesiveness of the community is considered. | 1 = Endemic (Conflict is frequent, is often violent, and is a pervasive aspect of daily life, e.g. feuding). <br> 2 = High (Conflict is frequent and often violent but is not a pervasive aspect of daily life.) <br> 3 = Moderate (Conflict occurs frequently but is seldom violent, or is violent but occurs only occasionally.) <br> 4 = Low (Conflict seldom occurs and is almost never violent.) | Order reversed |
| (No) internal warfare (between communities of the same society) | Warfare (i.e. lethal conflict between two or more groups of people) that takes place above the community level, but within the culture being coded. Thus, warfare between two villages belonging to the same culture, or between two political communities (consisting of multiple villages) belonging to the same culture. | 1 = Frequent, occurring at least yearly <br> 2 = Common, at least every five years <br> 3 = Occasional, at least every generation <br> 4 = Rare or never | Order reversed |
| (No) external warfare (with other societies) | Warfare (i.e. lethal conflict between two or more groups of people) between members of the culture being coded and any group that is not considered part of the culture being coded. | 1 = Frequent, occurring at least yearly <br> 2 = Common, at least every five years <br> 3 = Occasional, at least every generation <br> 4 = Rare or never | Order reversed |
| Estimate of culture population size | Population size | Number | Log-transformed |

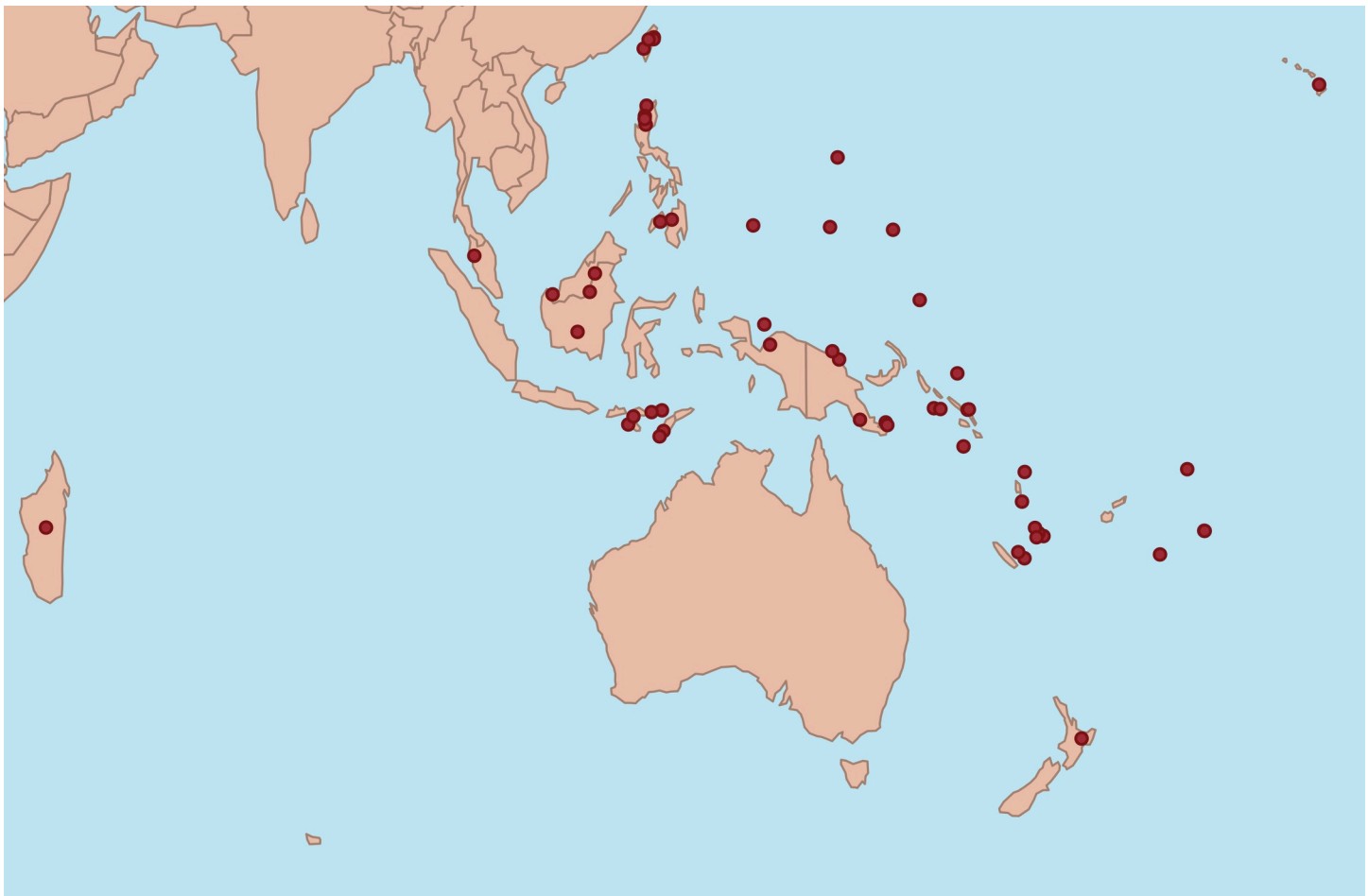

**Fig 1. Map indicating the approximate geographical location of the 54 Austronesian languages considered in our analyses.** Source: ww.naturalearthdata.com.

unreliable rate estimates, does not require divergence time estimates, and has higher statistical power to detect associations between rates of lexical change and fast-changing demographic or social parameters [54]. As Bromham [55] points out, selecting phylogenetically independent sister pairs is equivalent to running an experiment over and over again, taking one language, splitting it in two, and seeing which one evolves faster.

## Results

Multiple social, demographic and geographic factors exert independent effects on the rates of linguistic differentiation. For word gains, word losses as well as lexical turnover, the three full models including the five variables always provided the best fit, with a WAIC weight of 1 (S2–S4 Tables). The full models explained the majority of the proportion of the variance in rates of word gains, losses and overall lexical turnover (Bayesian $R^2$ values of 0.59, 0.70 and 0.66 respectively), also indicating that the five predictors had a stronger effect on rates of word losses.

Among the five factors, geographical isolation exerted the strongest effect on overall lexical differentiation, significantly increasing its pace (estimate = 1.09; 90% HPDI: [0.92, 1.24]) (Fig 2). This is because although it moderately increased the ability at which languages acquired new lexical items (estimate = 0.29, 90% HPDI: [0.06, 0.51]) it also severely increased the rate of word loss (estimate = 0.86, 90% HPDI: [0.70, 1.04]).

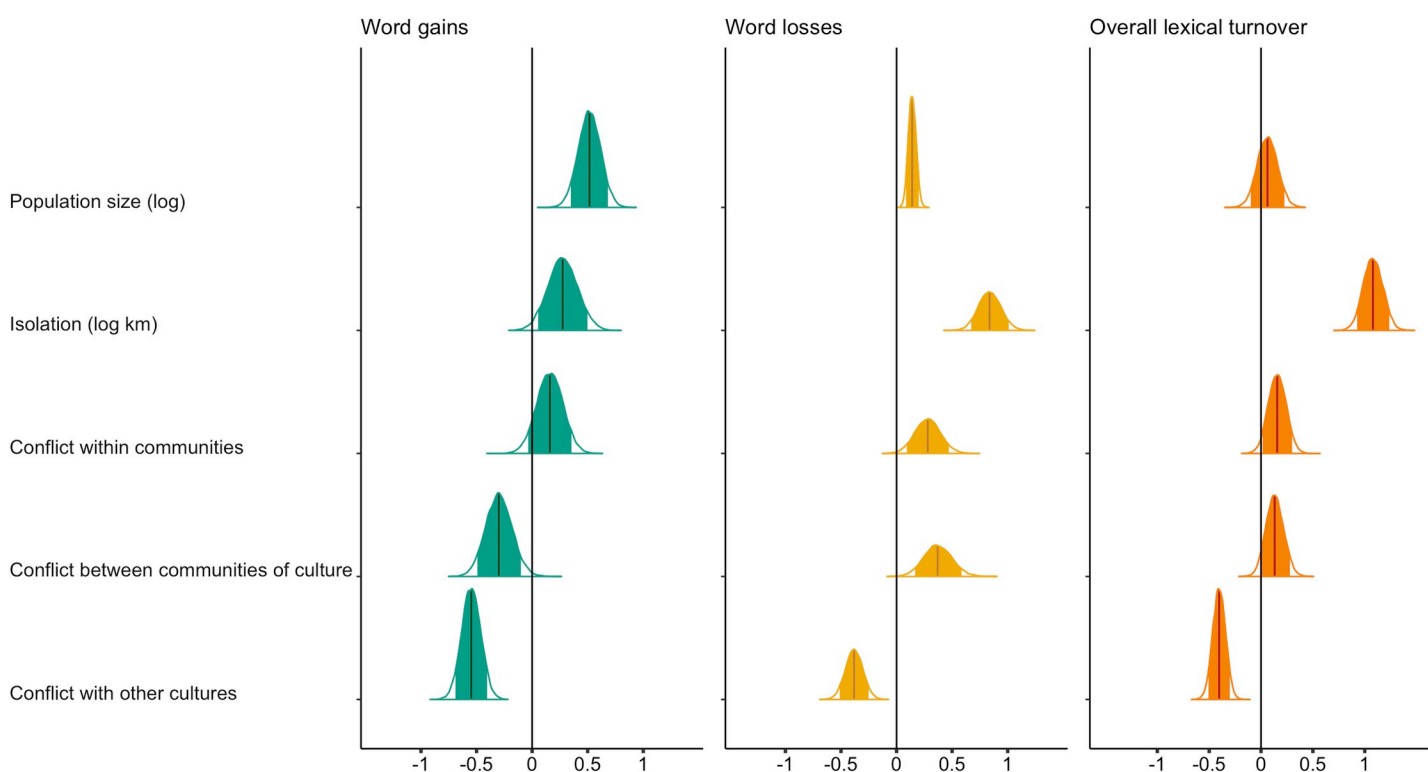

**Fig 2. Entire posterior distribution of the full models predicting the number of word gains (left), the number of word losses (centre) and the overall lexical turnover (word gains + word losses).** The thick line represents the median parameter estimate and the shaded region the 90% Highest Posterior Density Interval (HPDI) that reveals the narrowest interval containing the specified probability mass from 24,000 samples from the posterior distribution.

In the models, the languages of cultural groups with larger populations gained words at a faster rate (estimate = 0.52, 90% HPDI: [0.34, 0.68]). Although we also found evidence that these languages also lost words at a slightly faster pace (estimate = 0.14, 90% HPDI: [0.08, 0.20]), the overall positive effect of speaker population size on lexical turnover was not significant (0.06, 90% HPDI: [-0.10, 0.23])(Fig 1).

Even if within-community conflict did not have a significant effect on the rate of creation of new words (estimate = 0.16, 90% HPDI: [-0.03, 0.36]), it accelerated the rate of word loss (estimate = 0.27, 90% HPDI: [0.09, 0.47]). As a result, conflict within communities of the same culture had an overall significant positive effect on lexical differentiation (estimate = 0.16; 90% HPDI: [0.01, 0.16]).

Conflict between communities of the same culture reduced words gains (estimate = -0.30; 90% HPDI: [-0.50, -0.11]) but increased the pace at which languages lost words (estimate = 0.37; 90% HPDI: [0.16, 0.58]) (Fig 2). Consequently, there was no significant effect of conflict between communities of the same culture on the overall rate of lexical differentiation (estimate = 0.13; 90% HPDI: [-0.01, 0.27]). In contrast, inter-cultural conflict significantly reduced both the rate of word gains (estimate = -0.55; 90% HPDI: [-0.69, -0.40]) and word losses (estimate = -0.38; 90% HPDI: [-0.51, -0.25])(Fig 2), resulting in a deceleration in the overall rate of lexical differentiation (estimate = -0.40; 90% HPDI: [-0.51, -0.30]).

## Discussion

We applied a phylogenetic sister-pairs approach to three measures of lexical divergence among Austronesian languages and showed that the five sociodemographic factors were able to

explain most variation in lexical differentiation between Austronesian languages. Among the five variables, geographical isolation exhibited the largest effect on lexical turnover. We found that being geographically isolated (further away from a landmass inhabited by members of another culture) greatly speeded up the rate of word losses and also moderately that of word gains. This is consistent with the idea that geographical isolation results in a higher risk of random loss of cultural items due to incomplete inter-generational sampling of existing variation [17,19]. However, we also found evidence that new lexical innovations were more likely to be driven to fixation in more isolated cultures. A powerful effect of drift acting on cultural evolution has also been reported for other traits such as technological toolkits [23], phonemes [56] or even baby names across the United States [57].

We found no association between geographical isolation and within-community conflict in our sample ($\chi^2$ = 52.66, df = 44, p = 0.17; S2 Fig). Therefore it is unlikely that geographical isolation promotes differentiation either by increasing social group cohesion [33] or intensifying within-group conflict over resources [58]. In summary, we argue that geographical isolation may have contributed to differentiation among Austronesian languages mostly by preventing the random loss of existing lexical items, despite also hindering the ability of novel lexical innovations to emerge and be utilised [38,59], rather than by affecting the internal social dynamics of isolated groups.

Although population size did not have an effect on the overall rate of linguistic differentiation, larger population sizes increased the rate at which languages acquired new vocabulary items. Multiple studies have proposed that populations containing more individuals provide more chance for innovations to arise [19,60,61]. However, the positive association between population size and rates of word gains may also be due to large populations having less stringent norm enforcement allowing them to change faster [24,25]. At the same time, we also report a slight positive effect of population size on the rate at which languages lost existing lexical items. Whilst this is at odds with claims that smaller populations should be more prone word losses due to random sampling effects [16,17,19,23], influential linguistic theories, notably those by Lesley and James Milroy [26,62] have already remarked that small populations may have greater tolerance for diversity and malleable linguistic representations [63], facilitating their retention of linguistic variants. In contrast, we found no evidence for to smaller populations being able to gain new words more rapidly due to a faster uptake of innovations [25,64] nor for larger populations being less vulnerable to the loss of lexical items due to a larger number of sources of social learning and thus increased fidelity of information transmission [65].

Our findings are also at odds with Greenhill et al. [22], who did not find an association between population size and either rates of word gains or word losses in the Austronesian language family. We believe the reason for this discrepancy is twofold: First, they did not include any additional socioecological variables, whose confounding effects may have masked the effect of population size. When running our models including only population size as predictor, its effect decreased on the rate of word gains decreased from 0.52, 90% HPDI: [0.34, 0.68] to 0.09, 90% HPDI: [0.01, 0.18] and became insignificant for word losses (estimate = 0.01, 90% HPDI: [-0.10, 0.12]). But most importantly, the disparate results we observe are due to our choice of population size variable: whereas Greenhill et al. used contemporary speaker population sizes obtained from the Ethnologue [4], we used historical population sizes (prior to large-scale modernization trends taking place throughout past century). We believe that considering historical instead of contemporary population size estimates was important not only to match the temporal resolution of the other socioeconomic variables but also to establish a direction of causality. Since much of the lexical data used for the construction of the phylogenetic tree providing sister pairs for both our and Greenhill et al.'s analyses (see Materials and Methods) were obtained during the early twentieth century, contemporary speaker population sizes may

not be representative of the samples when linguistic data were obtained. For example, Hawaii is coded by the Ethnologue as having 2,000 speakers, despite it being one of the languages with the largest numbers of speakers historically [66]. In addition, if demographic variables were collected more recently than linguistic ones, population size would be more properly interpreted as an outcome (rather than a predictor or cause) of linguistic dynamics.

To verify that our discrepancy with Greenhill et al. [22] was due neither to our smaller sample size nor to our choice of statistical methods, we ran their generalized linear models as well as our Bayesian Poisson regressions on our own data, but using contemporary in-area speaker population sizes from the Ethnologue [4] as the only predictor. Both methods failed to identify a significant effect of contemporary speaker population size on either rates of word gains (estimate = 0.05, P = 0.12 with their method; estimate = 0.05, 90% HPDI: [-0.03, 0.13] with our method) or word losses (estimate = -0.02, P = 0.57; estimate = 0.01, 90% HPDI: [-0.07, 0.09]). Finally, although our sample is smaller, it was still three times as large as those utilised in similar previous studies [48], and importantly, more evenly representative of the different geographical areas where Austronesian languages are spoken as well as their different linguistic subgroups (Fig 2)

Our findings seemingly contradict those obtained in other cultural domains where well interconnected groups tend to lose cultural diversity faster, possibly due to the propensity of individuals to learn from successful cultural models and hence accelerated population convergence [67]. This was illustrated by the fact that high levels of intra-cultural conflict (both within communities and between communities of the same culture), which presumably reduces connectivity-within groups, accelerated word losses. A possible explanation for this phenomenon is that internal conflict may reduce the density and connectivity of local social networks [68] to a point where it hinders their ability to maintain social conventions including words [69,70]. In addition, within-group conflict may disrupt a well-documented positive feedback between social cohesiveness and the establishment of group-level traits such as language [71]. However, our finding was independent of population size, as the two variables were not significantly correlated (S2 Fig). The different conflict variables were not associated with each other either, hence we find no evidence for some groups being generally more prone to conflict than others.

Social isolation, which acts as a social barrier to cultural and linguistic exchange, had the opposite effect of geographical isolation of rates of linguistic differentiation. Conflict between cultures decelerated both the rate of word gains and losses. Inter-cultural conflict may discourage communication between speakers of different languages and therefore impose social (as opposed to physical) barriers to the emergence of new variants via the process of second language acquisition [72,73]. At the same time, since conflict poses a risk of acculturation or extinction, groups may turn to linguistic prescriptivism and the use of language as a marker of identity to prevent losses of structural integrity [45] and preserve group boundaries [41], thus resulting in a reduced rate of word loss.

Finally, although it is not within the scope of this study to provide a formal test on the role of cultural group selection on language evolution [46,47] our results did not support the idea that between-group conflict promotes linguistic differentiation at least in the case of Austronesian languages. In fact, Austronesia is famous for the importance of long-reaching networks of institutionalised ritualistic alliances such as the Kula Ring [74] and other cultural practices such as spousal exchanges, collective defence arrangements and exchange of social information and technology [75,76] aimed at creating ties between groups separated by thousands of kilometres. Those processes have been interpreted as adaptations for mitigating the potentially detrimental effects of isolation on the genetic and cultural diversity of insular populations [17]. Therefore, peaceful contact rather than warfare seems to contribute to extensive

multilingualism and fast lexical turnover among Austronesian populations, whose mean distance to the nearest landmass is over 128 km in our sample.

In summary, we identified increased isolation and internal conflict, and reduced between-group conflict, as factors contributing to linguistic differentiation. They operate by altering rates of word gain, word loss, or both. In addition, although population size did not have an effect on the overall rate of lexical evolution, languages spoken in larger communities gained words at a faster pace. Our conclusions may be specific to the case of Austronesian languages, spoken by populations separated by long distances and living on islands often unable to support multiple communities or cultural groups. It is not clear whether linguistic differentiation would reflect a different set of factors in groups where isolation does not pose an imminent risk, or on islands such as Papua New Guinea where a large number or groups are present and where social boundaries resulting from between-group conflict may be as relevant as geographical boundaries. In addition, the median number of speakers in our sample in only 26,485 compared to over seven million in Indo-European languages [4], which is relevant since the effect of population size may change beyond a given critical mass [77]. Finally, social and demographic factors may be more relevant to the evolution of lexicon than grammar or phonology, as the former have been shown to evolve in punctuational bursts rather than in a gradual fashion among Austronesian languages [39,40].

## Materials and methods

The Austronesian language family is one of the most diverse on the planet, comprising between 1,100 and 1,200 languages [78]. Austronesian cultures and languages are the product of a recent expansion [52] and thus Austronesian cultures and languages share many similarities with one another [79]. Their spread into Oceania was part of an expansion starting from Taiwan at around 5,000 ya, reaching the Bismarck Archipelago by 3,400 ya and Remote Oceania by 3,200 ya, in association with the appearance of Lapita pottery [80].

### Phylogenetic sister-pairs approach and selection of languages

To control for evolutionary relatedness between Austronesian languages we used the method of phylogenetically independent sister pairs [48,54]. We selected sister pairs that are each other's closest relatives, such that they share a more recent common ancestor with each other than with any other language in the sample. This implies that pairs are phylogenetically independent from each other [81], because any differences between the two languages in the same pair have evolved since their split from a common ancestor not shared with any other language in the sample (see Greenhill et al. [22] or Bromham et al. [48] for previous applications of the approach). This method allowed us to address the relationships between rates of word gains, losses and overall lexical turnover between sister languages on the one hand, and contrasts in five sociodemographic predictors on the other while controlling for phylogenetic ancestry. In other words, when two languages evolve from a common stock, our approach allowed us to investigate whether the more isolated language lose words at a faster rate than the one in closer contact to other languages?.

Phylogenetically independent pairs of languages were chosen from a previously published time-calibrated phylogenetic tree containing 400 Austronesian languages [52]. We trimmed the original phylogeny to include all languages that were also listed in the Pulotu dataset [82] covering the main Austronesian cultural groups. This left us with a new phylogeny composed of n = 86 languages. We then extracted sister pairs from the phylogeny, discarding any pairs whose classification was at the odds with the Ethnologue [4]. We also used phylogenetic support measures from published phylogenies as a guide to selecting well-attested sister pairs,

rejecting any pairs with less than 80% posterior probability in the published phylogeny. Last, we checked that the branch lengths between our sister pairs coincided with those reported by Greenhill et al. [22] and no disagreement was found. Furthermore, to reduce uncertainty, we excluded two sister pairs whose branch lengths were the entire tree (4,300 years) as they did not truly represent closely related languages but opposite ends of the phylogenetic tree. This left us with a final sample of 25 language pairs (n = 50 languages) representative of 11 major subgroups of the Austronesian language family (Polynesian, Micronesian, Southern Oceanic, South-East Solomonic, Papuan Tip, West New Guinea, Central Malayo-Polynesian, Western Malayo-Polynesian, Philippine, Meso-Melanesian and Central Pacific) (S1 Table; Fig 1).

The sister-pairs approach has two main advantages over whole tree phylogenetic methods that use every branch in a phylogeny as a datapoint in an analysis, namely: (i) using only the tips of the phylogeny avoids the need to infer less reliable ancestral states down the phylogeny, which is particularly important given that some Austronesian languages date back as far as 4,500 years ago; and (ii) using only tip branches also avoids the problem of non-independence between ancestor and descendant lineages within the phylogeny, as each branch is likely to be more similar in many traits to its immediate neighbours than to more distantly related branches simply due to relatedness [22].

## Social, demographic and geographic variables

All social, geographic and demographic data used for our analyses was obtained from the "traditional state" section in Pulotu database, which concerns the state of cultural groups prior to large-scale modernisation [82]. We made sure the ISO codes between the entries matched the taxa from our tree. Since the Pulotu database define a culture as "a group of people living in a similar physical, social and economic environment that speak mutually intelligible languages and have relatively homogenous supernatural beliefs and practices"[82], some of the cultures (24 out of 116) encompass speakers of different languages and were removed from our sample. Languages with insufficient linguistic, temporal or sociodemographic data were also excluded. This selection process resulted in 27 pairs (n = 54 languages) of Austronesian languages, although two of the pairs were removed from the final sample due to the reasons outlined above. Details on the variables extracted from the Pulotu dataset and our treatment prior to analysis are reported in Table 1. In addition, the Geographical Isolation variable ("Distance to the nearest landmass inhabited by a different culture") was verified using the "Distance Measurement Tool" function of Google Maps to estimate the shortest coast-to-coast distance. We found one disagreement (in the case of Hawaiian) so we modified our dataset to match the map-based estimates.

Before conducting our statistical analyses, we checked for multicollinearity among predictors using the generalized variance inflation factor (GVIF). All GVIF values fell below the lowest commonly recommended threshold of 2, indicating that our models should not suffer from multicollinearity (S6 Table) [83].

## Vocabulary data

We estimated rates of gain and loss of word variants [22,48]. We used the Austronesian Basic Vocabulary Database (ABVD) [78] which includes wordlists for 210 items of basic vocabulary ("basic semantic units") from over 500 Austronesian languages. Using basic vocabulary permits ensuring that cognate terms not only have a common history but a common meaning across language comparisons.

For each of the languages in our sample, we took each of the 210 identified basic vocabulary items as semantic units. For example, one semantic unit is "woman," which may be

represented by different words in different languages. The term "cognate set" represents a set of lexical units that are clearly related by descent and have been identified by linguists as being derived from a common ancestral word. For instance, many Polynesian languages share related terms for the word "five" such as "lima" in Fijian, "nima" in Tongan, "gima" in Rennellese, "'ima" in Marquesan, and "rima" in New Zealand Maori [78]. This represents a cognate set as the words descended from the ancestral the reconstructed protoform *lima in the ancestral Proto-Malayo-Polynesian language. Hence, when we say that a word in one language has a cognate in another language, we mean that both languages contain words from the same cognate set in the same semantic unit.

## Rates of language change

We identified patterns of word gain and loss by recording instances where a cognate form within a given semantic category was present in one language of a sister pair but not in the other[48,22]. If a word form found in one sister language has a cognate in other languages in the language family, it is likely to have been inherited from the common ancestor. This implies that the absence of that cognate form in the other sister language must be due to its loss after divergence from their exclusive common ancestor. On the other hand, if one of the sister languages has a unique word form with no recognised cognates in any other language in the family, it presumably represents a gain of a new word since the split from its sister language. Therefore, any changes in such terms between sister languages implies that they become more dissimilar to one another (i.e. have less words in common to define those 210 basic semantic units).

We did not include any identified loan words in the analysis, and therefore any cognate terms shared by two languages should be present due to inheritance from a common ancestor. This implies that the addition of a new word requires innovation as opposed to borrowing (horizontal transfer) from another language. Simulation studies of borrowing suggest that including loan words would make the sister languages seem more similar than they actually are by masking innovations or losses (see Greenhill et al. [84] for a discussion on this issue).

Since the addition of a new word does not necessarily involve the loss of an existing word (as languages can have multiple lexemes for one category), each recorded gain, or loss of a lexeme was counted as a separate event, regardless of semantic category. We did not consider cognate forms either present or absent in both members of a sister pair, as they provide no information on word gain or loss.

The total number of gains and losses were counted for all available semantic categories for each pair of languages using the Python package RateCounter developed by Simon Greenhill (https://github.com/SimonGreenhill/RateCounter; see [22] and S5 Table). The overall rate of lexical turnover was then computed for each language pair by adding the number of gains and losses.

## Statistical analysis

We used Bayesian inference for all statistical analyses. In a Bayesian framework, each model conditions data on prior probability distributions and uses Monte Carlo sampling methods to generate posterior distributions of estimated parameters. This framework allows us to compare entire posterior distributions, without relying on specific post-hoc tests and obviating the need to adjust for multiple comparisons. We are also better able to visualise and interpret differences between parameter estimates relative to a specific value, by reporting and displaying the entire posterior distribution for each predictor rather than assuming any particular threshold for statistical significance. In addition, standardisation of the variables allowed us to make direct comparisons of effect sizes.

We fitted three sets of Bayesian generalized linear models with Poisson link function, the first set predicting the rates of word gain, the second predicting the rate of word loss, and the third predicting the overall rate of lexical turnover. For each set, we fitted a null model (intercept only), a full or maximal additive model containing the set of five predictors (population size, geographical isolation, within-group conflict, between-group conflict (same cultural group), and between-group conflict (distinct cultural groups)), five models containing each predictor in isolation, and three additive models containing different combinations of the five predictors (S2–S4 Tables).

The difference in the number of word gains and losses between languages in each pair was modelled as a Poisson distribution, where the expected number of differences in words gained or lost is a log-linear function of the main effects β. We adopted regularising priors that are more conservative than the implied flat priors of non-Bayesian procedures, which prevents the model from overfitting data [85]. We have also fitted alternative model parameterisations, to verify that our results are qualitatively robust to changes in priors. The full Poisson additive model in each of the three sets was:

$$\log(\mu i) = \alpha + \beta_1 \log \text{Population} + \beta_2 \text{Conflict within communities} + \beta_3 \text{Isolation} + \beta_4 \text{Conflict within cultures} + \beta_5 \text{Conflict between cultures}$$

$$\alpha \text{ prior} \sim \text{Normal}(0, 10)$$

$$\beta \text{ prior} \sim \text{Normal}(0, 2)$$

Parameter estimation for each model was achieved with RStan [86], running three Hamiltonian Monte Carlo Markov chains in parallel until convergence was suggested by a high effective number of samples and $R^{\wedge}$ estimates of 1.00 [85]. This entailed 10,000 samples per chain, 2,000 of which were used as warm-up. We also visually inspected trace plots of the chains to ensure that they converged to the same target distribution and compared the posterior predictions to the raw data to ensure that the model corresponded to descriptive summaries of the samples. We also checked the bivariate correlations between all predictors (S2 Fig), none of which was significant.

For model comparisons, we used Widely Applicable Information Criteria (WAIC) which provides an approximation of the out-of-sample deviance that converges to the leave-one-out cross-validation approximation in a large sample. Analyses were performed in R 3.5.2 using the brms package [87,88]. We calculated model weights (the probability that a given model will perform best on new data) relative to other candidate models [85]. Recent extensions of the coefficient of determination $R^2$ generalised it to non-Gaussian distributions [89], which allows us to partition the proportion of variance captured by different predictors and evaluate their relative importance in explaining variation in rates of language change in our sample.

## Supporting information

**S1 Fig. Phylogenetic tree used to extract the sister pairs used in our analyses.** It is composed of all the languages included in Gray et al. [39] for which the Pulotu [69] database had an entry.
(DOCX)

**S2 Fig. Bivariate correlations between all the predictor variables included in the full model.** Numbers within the cells are Pearson's correlation coefficients. Blue cells are statistically significant (*p < 0.05)*, with bluer colors as *p* approaches zero; white cells are borderline

statistically significant; red and grey cells are not statistically significant.
(DOCX)

**S1 Table. Sister pairs included in our analysis.** The branch length indicates the number of years since the two languages diverged from a common ancestor, as obtained from *RateCounter* [11].
(DOCX)

**S2 Table. Model comparisons for word gains models.**
(DOCX)

**S3 Table. Model comparisons for word loss models.**
(DOCX)

**S4 Table. Model comparisons for lexical turnover models.**
(DOCX)

**S5 Table. Word gains, losses and overall lexical differences between the languages in each sister pair.**
(DOCX)

**S6 Table. GVIF values for each variable in the full models predicting word gains (top), word losses (middle) and overall lexical turnover (bottom).**
(DOCX)

**S1 Dataset. Sociodemographic and geographic predictor variables for each of the 54 languages.**
(CSV)

**S2 Dataset. Phylogenetic sister pairs with number of shared nodes and branch lengths.**
(CSV)

**S3 Dataset. Output from *RateCounter*, including word gains, losses and overall lexical turnover for each of the sister pairs.**
(CSV)

**S1 File. R Code to reproduce the analyses.**
(R)

## Acknowledgments

The authors wish to acknowledge Simon Greenhill for help with the implementation of Rate-Counter and Andrea Migliano for helpful discussions on previous versions of the paper. We also want to thank Terhi Honkola and two anonymous reviewers for their contribution to improving the manuscript.

## Author Contributions

**Conceptualization:** Cecilia Padilla-Iglesias, Lucio Vinicius.

**Data curation:** Cecilia Padilla-Iglesias, Erik Gjesfjeld.

**Formal analysis:** Cecilia Padilla-Iglesias, Erik Gjesfjeld.

**Investigation:** Cecilia Padilla-Iglesias.

**Methodology:** Lucio Vinicius.

**Supervision:** Lucio Vinicius.

**Validation:** Cecilia Padilla-Iglesias.

**Visualization:** Cecilia Padilla-Iglesias.

**Writing – original draft:** Cecilia Padilla-Iglesias, Lucio Vinicius.

**Writing – review & editing:** Cecilia Padilla-Iglesias, Erik Gjesfjeld, Lucio Vinicius.

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
