## [Decision Letter · Decision Letter 0]

17 Aug 2020

PONE-D-20-18721

Geographical and social isolation drive the evolution of Austronesian languages

PLOS ONE

Dear Dr. Padilla-Iglesias,

Thank you for submitting your manuscript to PLOS ONE. Again, sorry about the delay in handling your manuscript.

After careful consideration, we feel that it has merit but does not fully meet PLOS ONE’s publication criteria as it currently stands. Therefore, we invite you to submit a revised version of the manuscript that addresses the points raised during the review process.

Please consider carefully the suggestions provided by the reviewers, as well as their requests for clarification. I believe that most will improve your manuscript.

We look forward to receiving your revised manuscript.

Kind regards,

Francesc Calafell

Academic Editor

PLOS ONE

Journal Requirements:

2. We note that Figure 2 in your submission contain map images which may be copyrighted. All PLOS content is published under the Creative Commons Attribution License (CC BY 4.0), which means that the manuscript, images, and Supporting Information files will be freely available online, and any third party is permitted to access, download, copy, distribute, and use these materials in any way, even commercially, with proper attribution. For these reasons, we cannot publish previously copyrighted maps or satellite images created using proprietary data, such as Google software (Google Maps, Street View, and Earth). For more information, see our copyright guidelines: http://journals.plos.org/plosone/s/licenses-and-copyright.

2.1.    You may seek permission from the original copyright holder of Figure 2 to publish the content specifically under the CC BY 4.0 license. 

2.2.    If you are unable to obtain permission from the original copyright holder to publish these figures under the CC BY 4.0 license or if the copyright holder’s requirements are incompatible with the CC BY 4.0 license, please either i) remove the figure or ii) supply a replacement figure that complies with the CC BY 4.0 license. Please check copyright information on all replacement figures and update the figure caption with source information. If applicable, please specify in the figure caption text when a figure is similar but not identical to the original image and is therefore for illustrative purposes only.

Reviewers' comments:

Reviewer's Responses to Questions

**Comments to the Author**

1. Is the manuscript technically sound, and do the data support the conclusions?

Reviewer #1: Partly

Reviewer #2: Yes

Reviewer #3: Partly

2. Has the statistical analysis been performed appropriately and rigorously? 

Reviewer #1: No

Reviewer #2: Yes

Reviewer #3: Yes

3. Have the authors made all data underlying the findings in their manuscript fully available?

Reviewer #1: Yes

Reviewer #2: Yes

Reviewer #3: Yes

4. Is the manuscript presented in an intelligible fashion and written in standard English?

Reviewer #1: Yes

Reviewer #2: Yes

Reviewer #3: Yes

5. Review Comments to the Author

Reviewer #1: I have many questions/suggestions regarding the methods and how isolation scores were calculated that will need to be addressed before publication. The study is interesting and I would like to see it published, but these issues need to be addressed.

Manually check the calculated distances between cultures. Perhaps an expert in Pacific anthropology may be able to assist. I noticed the following errors:

Chamorro has no other cultures within 117km. The closest are over 600km away.

Hawaii is less than 1,900 km from Kiribati (1,700 km to the northern islands) but it is listed with an isolation of 3,335km

Sye is spoken on an island that had at one point four languages, which by the metrics used in this study, means that its isolation value should be 0.

Identify nearest cultures.

The reader needs to be given the location of the closest culture for each language, so that we can verify the numbers on isolation and replicate the study.

Reconsider 0-distance rating for languages on larger islands.

How can one justify listing a distance of 0 for cultures that are spoken on large islands? This is especially true for islands like Borneo or Madagascar (both islands are in the top 5 largest islands in the world). Distances between cultures may be far greater on these islands than between two smaller but closely situated islands. Listing them as 0 certainly skews results. Using islands as metrics in isolation distance only works in the part of the Austronesian world where islands are small enough to host single-language communities. The study incorrectly assumes that contact between two cultures located on different islands separated by a relatively small body of water will have less influence on one another than two cultures located at a great distance from one another on a single land mass. Austronesians are famously masters of over-ocean travel. There is no justification for treating 10km of water as isolated but 10km of land as non-isolated.

I worry that “culture” is a conflation of different factors, either one of which may impact the results. For example, two communities that speak the same language but follow different religious practices and two communities that speak different languages but have identical cultures, are both counted as “different cultures” but this hides the influence that language itself has on differentiation.

How does one come to the conclusion that “…contact has contributed to differentiation among Austronesian languages mostly by preventing the random loss of existing lexical items, despite also hindering the ability of novel lexical innovations to emerge and be utilized” when the most aberrant Austronesian languages, with incredibly low lexical retention rates and high levels of lexical turn over are found on New Guinea, an area where speakers have considerable contact with other cultures?

It is strange that two Loyalty Island languages are used, but languages of Mainland New Caledonia are not, even though Mainland New Caledonian languages have high lexical turnover rates and would have isolation values of 0.

Reviewer #2: The paper presents interesting findings and aspects about the evolution of Austronesian languages and deserves to be published. There are, however, several quite major issues in the paper that should be improved prior to publication. For that reason my recommendation is major revisions. Please find more specific comments below.

Introduction

The introduction needs major restructuring. More specifically my suggestions for the restructuring are the following:

It would be very important to clarify in the beginning what is the topic of study. In very general level it is “language evolution”, but as language evolution can cover different things, such as “language change”, “linguistic differentiation” and “language diversification” (terms which all are mentioned in the introduction) specificity is needed. This is important because “linguistic differentiation” and “language change” are not synonyms, but instead language change is part of linguistic differentiation process. That is, for the languages to differentiate or diverge - the process that has been the primary way of creating the 7000 languages - two components are needed: 1) language change (which is happening everywhere all the time) and 2) an isolating factor (geographical distance, biogeographical barriers, social factors etc.) (for more info and references see Honkola 2016 PhD thesis Micro and macroevolution of languages, section 1.4 and the reference to Foley 2004 within.) Thus, if you measure language change (as I understood you did in this study), you need to explain carefully what is the connection of it with linguistic differentiation, and whether you can make inferences about the latter based on the first.

Related on the point mentioned above, I would ask the authors to be specific and consistent in the usage of these terms throughout the manuscript as if those are not used in the specific manner, the reader may easily get confused. For example in the first paragraph you write “and may exhibit less stringent enforcement of norms, thus allowing languages to change faster. In contrast, other studies have argued that linguistic differentiation should be fastest in small populations due more rapid diffusion of new features--“ However, as language change and linguistic differentiation are different things (even though related, see above), their comparison is not very useful in the way it is done here. On the other hand if the comparison is correct and it is only the terms which are misleading, this should be fixed too. Similar kind of problem exists in the paragraph about geographical isolation. Please fix these.

In addition to the factors mentioned in the introduction (demographic, geographical isolation, warfare), it would be good to note (either in the introduction or in the discussion) that also the role of environment has been a topic of interest in relation to the emergence of language diversity and diversification in recent years (see e.g. Hua et al. 2019: The ecological drivers of variation in global language diversity. Pacheco Coelho et al. 2019: Drivers of geographical patterns of North American language diversity. Honkola et al. 2018: Evolution within a language: environmental differences contribute to divergence of dialect groups). On the other hand, if the topic is about language change and not linguistic differentiation (see comments above), then this is not necessary.

The end of the first paragraph of the Introduction (starting “In addition, most have used contemporary speaker population sizes in order --“) which mentions Austronesian family, need for the deeper time depth and problems in causality comes too early in the text as the introduction of different factors (geographical distance, warfare) continues after that. Please reorganize the text and introduce these points in the paragraph where you introduce the language family and the approach.

“Another factor often associated with linguistic diversification is warfare.” Please elaborate in which populations and where this association has been seen.

On my opinion the map should be Figure 1 as you could refer to that already in the Introduction when you introduce your sample of 54 languages. It is very relevant to show to the reader early on the location of the study populations.

Please introduce your three measures of three linguistic diversification (rates of gain and loss of basic vocabulary items, and their overall effect (rate of lexical turnover)) more thoroughly in the Introduction. This is crucial as this seems to be the link between language change and linguistic differentiation (see my comments above). E.g. How rates of gain and loss of basic vocabulary is connected to linguistic diversification? How do these measures differ from each other in relation to what they tell about linguistic diversification (i.e. is word gain more relevant than word loss, or is it the overall effect that matters the most)? Are they the common measures to use in this kind of a study/have they been used in other studies? This is very relevant as what you can say and infer about your results depends on this (whether you can just discuss about language change or can you make inferences about linguistic diversification).

I find the usage of the term “socio-ecological” misleading as no ecological variables were used in the analysis (this was used also in the Results section). Please change these.

It would be good to separate the short explanation of materials and methods in the Introduction into two paragraphs (one paragraph for materials and another for methods). In addition, as there is a separate Materials and methods section in the end of the paper, I would say that a lot of methodological details (basically the last two paragraphs of the Introduction) could go to the methods section there.

You refer to Table S1 in the last paragraph of the Introduction, but I believe this is wrong as Table S1 shows the sister pairs and not the different models that were analysed. Please fix this.

Results

I find it stylistically quite poor that each paragraph has its own section title. Please reconsider whether all of these are needed. Especially those results which are interpreted from Figure 1 should be within one section. Please also add references to the Figure 1 in all cases where it is relevant (e.g. also to “lexical innovation is faster in larger populations” section).

Would it be possible to have the Bayesian R2 values added to the Supplementary Information Tables?

In sections about the effect of population size and conflict between communities of the same culture it is said that the effect was not significant. In conflict within communities section you write “As a result, conflict within communities of the same culture had an overall positive effect on lexical differentiation.” Was this overall positive effect significant or not? Please be consistent in reporting these.

Discussion

In the current version the authors discuss the non-existence of association between geographical isolation and within-community conflict. I would think that it would be relevant to discuss also whether the different conflict-related variables were associated as it could tell us whether some groups are in general more prone to conflict than some others.

“Although population size did not have an effect on the overall rate of linguistic differentiation, larger

population sizes by increasing the rate at which languages acquired new vocabulary items.” Is there a word missing from this sentence? Please fix.

New analyses and results should not be presented in Discussion. Therefore I would suggest reorganizing the text in the way that the repetition of Greenhill et al.’s analyses are also introduced in the Methods and the results of that analysis in the Results section.

“To verify that our discrepancy with Greenhill et al.11 was due neither to our smaller sample size nor to

our choice of statistical methods, we ran the their generalized linear models” Is there something wrong in this sentence (the their)? Please fix.

“To verify that our discrepancy with Greenhill et al.11 was due neither to our smaller sample size nor to

our choice of statistical methods--” Above you mention that one another alternative for the discrepancy between your and Greenhill’s results could be that you used additional variables whereas Greenhill did not. Did you re-run the analyses with contemporary population sizes with additional variables too? This should be clarified. If this was not done yet, perhaps it would be worth doing?

Materials and methods

On my opinion, thorough restructuring is needed in this section too as I find the current order of paragraphs confusing. General and specific points concerning this section are:

“Austronesian cultures and languages are the product of a recent expansion and thus

share many features, and are largely found in similar environments.” To me it seems like a bold claim to say that environmental conditions are similar considering the size of the territory (thousands of kilometres) in which the studied Austronesian languages are spoken. The point about similarity of the environments should be specified or clarified.

Please separate materials and methods sections from each other. That is, first, list all materials you were using. Also, refer to Table 1 also in the Materials and methods section. Only after this explain about your methods. I understand that sister-pair method is related to the selection of your data and you can say that this method was guiding your data selection, but please explain the details of that method only after you have explained the datasets.

I would also like to have more details about your variables. “We selected all the languages that were also listed in the Pulotu dataset covering the main Austronesian cultural groups. “ How many languages is this? Were the sister pair picked by hand? It would also be good the sister pairs marked to the Fig. S1.

In relation to Fig S1, you write in the legend of Fig S1 “Phylogenetic tree used to extract the sister pairs used in our analyses. It is composed of all the languages included in Gray et al.39 for which the Pulotu69 database had an entry.” So did you make this phylogeny specifically for this study? If yes, this should be clarified and explained in the methods section (how it was done etc.).

“Last, we checked that the branch length between our sister pairs coincided with those reported by Greenhill et al.” ‘Please explain why this was done.

Please elaborate why three different conflict-related variables were chosen, and not just one of them. In addition, I find it surprising that there was no multicollinearity between these different conflict-factors. It would be good to have the GVIF table as a supplementary table.

In the section Rates of language change you talk about “semantic categories”. Is this a different thing that “meaning” which is the term commonly used with basic vocabulary lists. In general, this comment relates to the fact that currently too little is said about the linguistic basic vocabulary data used in this study.

Similarly, in the following paragraph you write “word does not necessarily involve the loss of an existing word”. Do you also here refer to “meaning” when you write “word”. Please be specific and consistent with these terms too.

“We did not include any identified loan words in the analysis, and therefore any cognate terms shared

by two languages should be present due to inheritance from a common ancestor. This implies that the

addition of a new word requires innovation as opposed to borrowing (horizontal transfer) from another

language.” As far as I can understand, these two sentences are not in line with each other. What if the borrowing arises individually to one of the sister-pair languages and not via the common ancestor, that is, it looks like an innovation but it is borrowed. Were those also excluded? Please clarify.

“To reduce uncertainty, we excluded two sister pairs whose branch lengths were the

entire tree (4,300 years) as they did not truly represent closely related languages but opposite ends of

the phylogenetic tree.” Does this mean you actually had 25 pairs (and 50 languages) in total in your study? In both cases (whether you had 27 or 25 pairs) this should be written already when you introduce your datasets/variables.

In the section Statistical analyses, please say clearly again that it a regression analysis what you are using here and what were the dependent and independent variables. This was currently in the last paragraph of the Introduction but on my opinion it should be in the Statistical analyses section too.

Were the “conflict-variables” sometimes analysed jointly in the different combination models or why is it marked only as “Conflict” in the SI Tables 2-4. This is a critical issue as for example you really cannot say that you analysed five variables if you in reality only analysed three of them.

Figures

On my opinion Figure 2 with the map covers an unnecessarily large geographical area (nearly half of the map area does not have data collection points). To save space this could be made notably smaller.

Reviewer #3: The paper was written in a dense format. Many details are not spelled out and the paper needs revision that will improve readability. As of now, for a non-Bayesian non-phylogenetic reader, the paper is tough to understand. I would strongly suggest that the authors work on this aspect. The paper has merit but the sample size is not convincing enough and requires more experiments.

My comments are below.

Recent models have added population

structure as an essential demographic factor underlying cultural and linguistic evolution. For example,

population density, local interconnectedness and migrations were claimed to play a key role in

cumulative cultural evolution by facilitating the emergence, diffusion and survival of linguistic

innovations 7,10,17,18,19,20 .

Can the references be unbundled so that it makes more sense? Does the referencing follow PLOS ONE format?

ongoing controversy over  I am not aware of this controversy. I think of this as an ongoing debate rather than a controversy.

using only tip branches.  What is tip branch?

Are the 27 language pairs representative of the major subgroups (more than ten of them) of Austronesian? Adding this information would be really useful for the reader.

Monte Carlo sampling method  Monte Carlo Markov Chain sampling method

Poisson link function  Is it a Poisson regression? (generalized linear model). Then, the MCMC sampling is just another way to do the estimation which might be useful in the case of small data.

R^ estimates  What is the statistic that is being referred to?

How are the word losses and gains estimated? is it from an ancestral construction analysis?

How is lexical turnover being computed? Sum of gain and loss or is it estimate from the data?

Can the regression be run for more languages only with two factors: geographical isolation and population size? The small number of languages could make a big difference in the interpretations. Also, the languages' sample coverage might contribute to the contradictory results with Greenhill et al. If it is a larger dataset, then, the regression without the conflict variables might support what Greenhill et al. found.

6. PLOS authors have the option to publish the peer review history of their article (what does this mean?). If published, this will include your full peer review and any attached files.

Reviewer #1: No

Reviewer #2: **Yes: **Terhi Honkola

Reviewer #3: No

---

## [Author Response · Author response to Decision Letter 0]

3 Oct 2020

Dear Dr Calafell

We thank you for editing our manuscript and giving us the opportunity to revised it. We believe we have addressed all questions raised by the reviewers and hope you are satisfied with the replies and the new manuscript. 

First of all, regarding the issues related to the formatting of the manuscript, we have now re-formatted it according to PLOS ONE’s style requirements. We have also re-made the Figure 2 using a map from Natural Earth Data (ww.naturalearthdata.com), which is free of copyright and therefore suitable to be made freely available online for anyone to access, download, copy, distribute, and use these materials in any way.

Secondly, in the light of the reviewers more specific comments, we have amended the text as shown below.

Reviewer 1:

Re. Manually check the calculated distances between cultures. Perhaps an expert in Pacific anthropology may be able to assist. I noticed the following errors:

Chamorro has no other cultures within 117km. The closest are over 600km away.

Hawaii is less than 1,900 km from Kiribati (1,700 km to the northern islands) but it is listed with an isolation of 3,335km

Sye is spoken on an island that had at one point four languages, which by the metrics used in this study, means that its isolation value should be 0.

The variable was extracted from v.3 in the Pulotu database. The codebook that comes with it, available as supplementary material in Watts et al. (2015) defines it as: A different culture, for the purposes of this coding sheet, is any culture other than the culture being coded. The distance stated should be a minimum distance (see Watts et al. 2015 – reference 69 in the main text; and e.g. Kline and Boyd, 2010; Bromham et al. 2015; Nettle, 1999; Atkinson, 2011; Huisman et al. 2019). For example, if the culture being coded occupied an archipelago, and the nearest other culture occupied an archipelago located to directly to the east, the distance given should be the distance from the easternmost island of the western archipelago to the westernmost island of the eastern archipelago: If there was a different culture living on the same island, we coded this distance as “0”. 

We have also now manually checked all the distances reported – and verified the validity of the data from Pulotu. The Chamorro have the Carolinians (another Austronesian group comprising about 3100 people) within 117km of their reach. 

As for the distance between Hawaii and Kiribati – it is disputed whether Napari (the northernmost island from Kiribati, located indeed 1759km from Hawaii) had any inhabitants at all. In 2010, 194 people were recorded (http://www.climate.gov.ki/wp-content/uploads/2013/01/21_TABUAERAN-revised-2012.pdf). We have calculated the distance between the southernmost coast of Hawaii (in Hilo island) and the second northernmost island from Kiribati is of 2117km. On a log scale, both these values would become 7.47 and 7.66 (with the values used in the previous version of the manuscript this value was of 8.11). We have amended the dataset and re-ran the analysis and there has been no qualitative changes (some of the point estimates have fluctuated by 0.1, but this can also be due to stochasticity in the MCMC sampling process from the posterior distribution). The rest of the distances do match what we have observed.

Regarding the last point, please notice that following the definition of culture we applied in our study, the fact that in the island where Sye is spoken another 4 languages were spoken does not mean that such island was inhabited by four different cultures. For example, the Buka people (a cultural unit not included in the present analysis) comprises speakers of at least 5 different but related Austronesian languages.

Re. Identify nearest cultures.The reader needs to be given the location of the closest culture for each language, so that we can verify the numbers on isolation and replicate the study.

Our data come from a peer-reviewed database also in PLOS ONE (again see reference 69 in the main text). All data are available to the public. Based on our evaluations, we have no reason to believe the published dataset is problematic. It is also freely accessible following this link: https://pulotu.shh.mpg.de

Re. Reconsider 0-distance rating for languages on larger islands.How can one justify listing a distance of 0 for cultures that are spoken on large islands? This is especially true for islands like Borneo or Madagascar (both islands are in the top 5 largest islands in the world). Distances between cultures may be far greater on these islands than between two smaller but closely situated islands. Listing them as 0 certainly skews results. Using islands as metrics in isolation distance only works in the part of the Austronesian world where islands are small enough to host single-language communities. The study incorrectly assumes that contact between two cultures located on different islands separated by a relatively small body of water will have less influence on one another than two cultures located at a great distance from one another on a single land mass. Austronesians are famously masters of over-ocean travel. There is no justification for treating 10km of water as isolated but 10km of land as non-isolated.

Evidence from previous studies suggests that in precontact times, these islands were at carrying capacity (Bromham et al. 2015; Kirch and Rallu, 2007). Most cultural groups that are now encapsulated are so as a result of post-colonial urbanization or globalization practices – and therefore such isolation post-dates the rest of the data we have used for this analyses. Hence, the most likely event is that languages in large islands wouldn’t have been isolated from one another. Furthermore, a 0 rating was only assigned if the distance to the nearest landmass inhabited by another culture was 0. Island size and pre-contact speaker population size in these area are also strongly correlated, so analysing both variables together would have been unjustified (Bromham et al. 2015). In summary, the approach we adopted was not invented by ourselves but follows along the lines of previous studies (Bromham et al. 2015; Kirch and Rallu, 2007; Kline and Boyd, 2010; Atkinson, 2011; Huisman et al. 2019). 

Re. I worry that “culture” is a conflation of different factors, either one of which may impact the results. For example, two communities that speak the same language but follow different religious practices and two communities that speak different languages but have identical cultures, are both counted as “different cultures” but this hides the influence that language itself has on differentiation.

Again, our definition and categorization of cultures corresponds to those made by Watts et al. 2015, where the word “culture” is used in two closely related ways. The first meaning (“culture”, with no article) denotes traditions (beliefs, practices and knowledge) that are transmitted by social learning (Currie, Greenhill & Mace, 2010, p 3904; Mace & Jordan, 2005, p 116). The second and more specific meaning (“a culture”) denotes the set of traditions that are characteristic of a particular group of people, or, conversely, a group of people characterised by a particular set of traditions.

Important to note that since we wanted cultures to match ethnolinguistic groups, we removed those cultural groups that were linguistically heterogeneous. In other words, we only selected cultures that had their own characteristic associated language.

Re. How does one come to the conclusion that “…contact has contributed to differentiation among Austronesian languages mostly by preventing the random loss of existing lexical items, despite also hindering the ability of novel lexical innovations to emerge and be utilized” when the most aberrant Austronesian languages, with incredibly low lexical retention rates and high levels of lexical turn over are found on New Guinea, an area where speakers have considerable contact with other cultures?

It is strange that two Loyalty Island languages are used, but languages of Mainland New Caledonia are not, even though Mainland New Caledonian languages have high lexical turnover rates and would have isolation values of 0.

The conclusion above was strictly derived from our results. The conclusion simply make explicit the findings on rates of word loss, word gain, and overall lexical turnover. As is known to anyone familiar with quantitative methods, their purpose is to derive general patterns. Obviously, there may be exceptions, but they do not invalidate the approach. 

However, we did not cherry pick languages to include in our sample but took ALL those languages from Gray et al.’s phylogenetic tree for which there was an entry in Pulotu (and thus, available sociodemographic data). 

We thank the reviewer for the comments, but in our view our study relies on solid, peer-reviewed and frequently used methodology and databases.

#########################################################################

Reviewer 2:

Re. It would be very important to clarify in the beginning what is the topic of study. In very general level it is “language evolution”, but as language evolution can cover different things, such as “language change”, “linguistic differentiation” and “language diversification” (terms which all are mentioned in the introduction) specificity is needed. This is important because “linguistic differentiation” and “language change” are not synonyms, but instead language change is part of linguistic differentiation process. That is, for the languages to differentiate or diverge - the process that has been the primary way of creating the 7000 languages - two components are needed: 1) language change (which is happening everywhere all the time) and 2) an isolating factor (geographical distance, biogeographical barriers, social factors etc.) (for more info and references see Honkola 2016 PhD thesis Micro and macroevolution of languages, section 1.4 and the reference to Foley 2004 within.) Thus, if you measure language change (as I understood you did in this study), you need to explain carefully what is the connection of it with linguistic differentiation, and whether you can make inferences about the latter based on the first

First of all – thank you for sharing your PhD thesis, it has proven extremely thought-provoking. Nevertheless, I believe the point Rob Foley was trying to make regarding language change and language differentiation (and with which I agree) was about the divergence of single languages. His point was that within a population language will always change, just ‘drifting’ along. When parts of that population become less inter intelligible, then language differentiation has or is occurring. In many cases, rather like with sub-species, they may merge back together again, and there may be issues of how to distinguish between increased variability within a group of language speakers, and differentiation, but that’s what it is! Within such context, his point was that isolation between speakers that used to belong to a single population is not necessarily geographical, although I would assume for most of the time in small scale societies it would be. 

The starting point in our analysis are already distinct languages, and we are considering change in different directions between languages already spoken in distinct populations. Especially since we are considering the gain and loss of cognates (analogous terms) in our study, “change” is fundamentally diversification – as a “gain” means that a language has a new word not found in her sister language and a loss means that a language has lost a word that used to share with her sister language. In other words – all change makes sister languages more dissimilar to one another. 

In other words, the kind of language change that we are measuring here is a directional change by which languages become more dissimilar to one another and therefore, in this case, the lexical change we are measuring becomes inevitably language differentiation. 

Re. Related on the point mentioned above, I would ask the authors to be specific and consistent in the usage of these terms throughout the manuscript as if those are not used in the specific manner, the reader may easily get confused. For example in the first paragraph you write “and may exhibit less stringent enforcement of norms, thus allowing languages to change faster. In contrast, other studies have argued that linguistic differentiation should be fastest in small populations due more rapid diffusion of new features--“ However, as language change and linguistic differentiation are different things (even though related, see above), their comparison is not very useful in the way it is done here. On the other hand if the comparison is correct and it is only the terms which are misleading, this should be fixed too. Similar kind of problem exists in the paragraph about geographical isolation. Please fix these.

Please see our reply above. Given the connection the article establishes between the two concepts, we do not believe that the terminology we use is confusing. 

For example, if we take the sentences quoted above, ultimately they mean:

“and may exhibit less stringent enforcement of norms, thus allowing languages to change [and therefore also differentiation between languages to occur] faster. In contrast, other studies have argued that [language change, and therefore] linguistic differentiation should be fastest in small populations due more rapid diffusion of new features--

Re. In addition to the factors mentioned in the introduction (demographic, geographical isolation, warfare), it would be good to note (either in the introduction or in the discussion) that also the role of environment has been a topic of interest in relation to the emergence of language diversity and diversification in recent years (see e.g. Hua et al. 2019: The ecological drivers of variation in global language diversity. Pacheco Coelho et al. 2019: Drivers of geographical patterns of North American language diversity. Honkola et al. 2018: Evolution within a language: environmental differences contribute to divergence of dialect groups). On the other hand, if the topic is about language change and not linguistic differentiation (see comments above), then this is not necessary.

Of course, we are aware of these factors and recognise their importance, nonetheless, exploring and discussing them goes beyond the scope of the present work. However, see acknowledgement of these issues in page 2 of the introduction.

Re. The end of the first paragraph of the Introduction (starting “In addition, most have used contemporary speaker population sizes in order --“) which mentions Austronesian family, need for the deeper time depth and problems in causality comes too early in the text as the introduction of different factors (geographical distance, warfare) continues after that. Please reorganize the text and introduce these points in the paragraph where you introduce the language family and the approach.

This has been amended accordingly. Thanks for the suggestion.

Re. “Another factor often associated with linguistic diversification is warfare.” Please elaborate in which populations and where this association has been seen.

We have added an explanation of why this might be the case and given some ethnographic and historical examples in the introduction.

Re. On my opinion the map should be Figure 1 as you could refer to that already in the Introduction when you introduce your sample of 54 languages. It is very relevant to show to the reader early on the location of the study populations.

The order of the figures has been amended as suggested. Thanks.

Re. Please introduce your three measures of three linguistic diversification (rates of gain and loss of basic vocabulary items, and their overall effect (rate of lexical turnover)) more thoroughly in the Introduction. This is crucial as this seems to be the link between language change and linguistic differentiation (see my comments above). E.g. How rates of gain and loss of basic vocabulary is connected to linguistic diversification? How do these measures differ from each other in relation to what they tell about linguistic diversification (i.e. is word gain more relevant than word loss, or is it the overall effect that matters the most)? Are they the common measures to use in this kind of a study/have they been used in other studies? This is very relevant as what you can say and infer about your results depends on this (whether you can just discuss about language change or can you make inferences about linguistic diversification).

We have now done so, see introduction (page 3) – for a more detailed explanation of these measures.

Re. I find the usage of the term “socio-ecological” misleading as no ecological variables were used in the analysis (this was used also in the Results section). Please change these.

These have been removed and replaced with socio-demographic. 

Re. It would be good to separate the short explanation of materials and methods in the Introduction into two paragraphs (one paragraph for materials and another for methods). In addition, as there is a separate Materials and methods section in the end of the paper, I would say that a lot of methodological details (basically the last two paragraphs of the Introduction) could go to the methods section there.

Although we feel one of those paragraphs is an important contribution of the paper to the existing literature, we have re-structured this section and moved the other paragraph to the Methods section as suggested.

Re. You refer to Table S1 in the last paragraph of the Introduction, but I believe this is wrong as Table S1 shows the sister pairs and not the different models that were analysed. Please fix this.

Indeed this is a mistake, and has been corrected. See the new Methods section, as that paragraph has been removed from the introduction.

Re. I find it stylistically quite poor that each paragraph has its own section title. Please reconsider whether all of these are needed. Especially those results which are interpreted from Figure 1 should be within one section. Please also add references to the Figure 1 in all cases where it is relevant (e.g. also to “lexical innovation is faster in larger populations” section).

The section headers have been removed, and now the Results section is written as a single text block.

Re. Would it be possible to have the Bayesian R2 values added to the Supplementary Information Tables?

Bayesian R2 define the proportion of the variance explained by the model. These should only be used after model selection, as the non-selected models could explain more variance yet be overfitted or due to confounds. For example, we could build a model with an arbitrarily large number of predictors so that 100% of the variance was explained. Such model would be heavily penalized by WAIC, obtaining a weight of 0, yet have a Bayesian R2 of 1. Hence, only when we have selected the model we can use, we should determine the proportion of the variance explained by it. Therefore, the three Bayesian R2 values we obtained from the three best models are reported in the main text.

Re. In sections about the effect of population size and conflict between communities of the same culture it is said that the effect was not significant. In conflict within communities section you write “As a result, conflict within communities of the same culture had an overall positive effect on lexical differentiation.” Was this overall positive effect significant or not? Please be consistent in reporting these.

Please see amended paragraph.

Re. In the current version the authors discuss the non-existence of association between geographical isolation and within-community conflict. I would think that it would be relevant to discuss also whether the different conflict-related variables were associated as it could tell us whether some groups are in general more prone to conflict than some others.

See page 5 for such discussion.

Re. “Although population size did not have an effect on the overall rate of linguistic differentiation, larger population sizes by increasing the rate at which languages acquired new vocabulary items.” Is there a word missing from this sentence? Please fix.

This has been fixed. Thank you very much.

Re. “To verify that our discrepancy with Greenhill et al.11 was due neither to our smaller sample size nor to our choice of statistical methods, we ran the their generalized linear models” Is there something wrong in this sentence (the their)? Please fix.

The wording of the sentence has been fixed.

Re. “To verify that our discrepancy with Greenhill et al.11 was due neither to our smaller sample size nor to our choice of statistical methods--” Above you mention that one another alternative for the discrepancy between your and Greenhill’s results could be that you used additional variables whereas Greenhill did not. Did you re-run the analyses with contemporary population sizes with additional variables too? This should be clarified. If this was not done yet, perhaps it would be worth doing?

We do not think this would be justified. Since we are arguing that their choice of population size variable is not justified – and therefore not evolutionarily representative, we don’t feel like it would be of any good to science to re-run the same analyses with the same incorrect variable. 

Re. “Austronesian cultures and languages are the product of a recent expansion and thus

share many features, and are largely found in similar environments.” To me it seems like a bold claim to say that environmental conditions are similar considering the size of the territory (thousands of kilometres) in which the studied Austronesian languages are spoken. The point about similarity of the environments should be specified or clarified.

To avoid any confusion, this point has been removed, instead, we have added a reference to Pawley’s 1967 work discussing the similarities between Austronesian cultures and how these are related to the fact that they are the product of a recent agricultural expansion.

Re. Please separate materials and methods sections from each other. That is, first, list all materials you were using. Also, refer to Table 1 also in the Materials and methods section. Only after this explain about your methods. I understand that sister-pair method is related to the selection of your data and you can say that this method was guiding your data selection, but please explain the details of that method only after you have explained the datasets.

We have tried to do it as much as possible. However, the phylogenetic sister pairs approach is performed in order to obtain the materials (i.e. the languages) used for the analysis. Apart from that, we separated materials from methods. 

Re. I would also like to have more details about your variables. “We selected all the languages that were also listed in the Pulotu dataset covering the main Austronesian cultural groups. “ How many languages is this? Were the sister pair picked by hand? It would also be good the sister pairs marked to the Fig. S1.

No, they were not picked by hand: We trimmed the phylogeny to include all languages that were also listed in the Pulotu dataset (see reference 69 in the main text) covering the main Austronesian cultural groups. We then extracted sister pairs from the phylogeny (in other words, all pairs of languages that were more closely related to one another, than to any other language in the phylogeny), discarding any pairs whose classification was at the odds with the Ethnologue (Reference 4 in the main text). We took the original file and trimmed the tree to contain only the languages in Pulotu. The sister pairs are clearly identified in Table S1 and their divergence times indicated.

Re. In relation to Fig S1, you write in the legend of Fig S1 “Phylogenetic tree used to extract the sister pairs used in our analyses. It is composed of all the languages included in Gray et al.39 for which the Pulotu69 database had an entry.” So did you make this phylogeny specifically for this study? If yes, this should be clarified and explained in the methods section (how it was done etc.).

See new description. It is not a new phylogeny. We took original files from Gray et al, whose timings have been validated multiple times both statistically and with archaeological and linguistic proxies and cut it to only include the languages in the Pulotu dataset – but we didn’t rebuild the tree (i.e. use any additional statistical methods to re-estimate divergence times or branch lengths ourselves) – that’s why branch lengths and all the rest remain the same.

Re. “Last, we checked that the branch length between our sister pairs coincided with those reported by Greenhill et al.” ‘Please explain why this was done.

Simply a housekeeping check to make sure after trimming the tree, the lengths between the remaining branches remained the same. Thus, it was done as described in the sentence. In other words, simple as a validation check.

Re. Please elaborate why three different conflict-related variables were chosen, and not just one of them. In addition, I find it surprising that there was no multicollinearity between these different conflict-factors. It would be good to have the GVIF table as a supplementary table.

Regarding multicollinearity – please check the correlation coefficients between the variables in the supplementary material, and verify that none of them were significantly correlated (Fig. S2). Additionally, although the code to compute them had been included as SM as well, we have now attached a table with the GVIF values for each model 

Re. In the section Rates of language change you talk about “semantic categories”. Is this a different thing that “meaning” which is the term commonly used with basic vocabulary lists. In general, this comment relates to the fact that currently too little is said about the linguistic basic vocabulary data used in this study.

Please see the new “Vocabulary data” section for a detailed explanation of this issue and the one below.

Re. Similarly, in the following paragraph you write “word does not necessarily involve the loss of an existing word”. Do you also here refer to “meaning” when you write “word”. Please be specific and consistent with these terms too.

“We did not include any identified loan words in the analysis, and therefore any cognate terms shared

by two languages should be present due to inheritance from a common ancestor. This implies that the

addition of a new word requires innovation as opposed to borrowing (horizontal transfer) from another

language.” As far as I can understand, these two sentences are not in line with each other. What if the borrowing arises individually to one of the sister-pair languages and not via the common ancestor, that is, it looks like an innovation but it is borrowed. Were those also excluded? Please clarify.

Thanks for the comments. We hope that the new paragraph explaining what we (and other authors) mean by “cognates” will help clarify this issue. Since a shared cognate between the two languages in a pair represents lexical units that are related by descent (derived from a common ancestral word), the purpose of removing loan words is to make sure that a shared word between two languages cannot be due to both of them borrowing the same word from a third language. For example, if both Gaddang and Isnag (one of our sister pairs) borrowed the word “okey” from English – this “shared word” would not represent a shared cognate (as it is not shared due to their common descent), and therefore would not be counted. Similarly, if only Gaddang borrowed the word “okay” from English, this would also not be counted as a word gain. Loan words were removed from database to discount the possibility that the cognate was lost from the common ancestor and then regained in one of the two languages. Simulation studies of borrowing suggest that including loan words would make the sister languages seem more similar than they actually are by masking innovations or losses (see Greenhill et al. 2009 for a discussion on the issue of horizontal transmission in cultural phylogenies). 

A shorter version of the explanation above has been added to the main text (see page 7).

Re. “To reduce uncertainty, we excluded two sister pairs whose branch lengths were the

entire tree (4,300 years) as they did not truly represent closely related languages but opposite ends of

the phylogenetic tree.” Does this mean you actually had 25 pairs (and 50 languages) in total in your study? In both cases (whether you had 27 or 25 pairs) this should be written already when you introduce your datasets/variables.

Thank you for pointing this out. The removal of those two “pseudo-pairs” was a modification done at a later stage as suggested by Simon Greenhill and Erik Gjesfjeld (the second author) in order to increase the robusticity of the method, and I (the first author) forgot to modify the rest of the manuscript accordingly. This has now been amended.

Re. In the section Statistical analyses, please say clearly again that it a regression analysis what you are using here and what were the dependent and independent variables. This was currently in the last paragraph of the Introduction but on my opinion it should be in the Statistical analyses section too.

Thank you for pointing this out. To increase precision, both in the introduction and in the methods section, we have included the term “Poisson Generalized Linear Model”.

Re. Were the “conflict-variables” sometimes analysed jointly in the different combination models or why is it marked only as “Conflict” in the SI Tables 2-4. This is a critical issue as for example you really cannot say that you analysed five variables if you in reality only analysed three of them.

The full (or maximal additive) models, which were the best-performing ones in all cases, included population size, geographical isolation and the three conflict variables. In tables S2-S4, you can see that in order to perform the model comparisons and selection, we fitted each variable separately as well as different combinations of them, as is the standard procedure – see Vehtari et al. 2017 or McElreath, 2015.

Re. On my opinion Figure 2 with the map covers an unnecessarily large geographical area (nearly half of the map area does not have data collection points). To save space this could be made notably smaller.

See the new version of Figure 2.

We would like to thank the reviewer for the very useful and detailed comments. We hope our replies and changes in the main manuscript have clarified the raised issues. 

#########################################################################

Reviewer 3: 

Re. The paper was written in a dense format. Many details are not spelled out and the paper needs revision that will improve readability. As of now, for a non-Bayesian non-phylogenetic reader, the paper is tough to understand. I would strongly suggest that the authors work on this aspect. The paper has merit but the sample size is not convincing enough and requires more experiments.

We thank the reviewer for the comments. Our new manuscript was rewritten based on comments from all reviewers, and although we believe the main points had been made clearly, we believe the suggestions increased the article readability. Once again, we wish to reiterate that our sample size is 2.5x that of Greenhill’s smaller study, as well as included not only Polynesian languages but languages from diverse Austronesian subfamilies. 

Re. Can the references be unbundled so that it makes more sense? Does the referencing follow PLOS ONE format?

Yes – this has been resolved.

Re. ongoing controversy over  I am not aware of this controversy. I think of this as an ongoing debate rather than a controversy.

This has been amended accordingly.

Re. using only tip branches.  What is tip branch?

The tip of the branch of the phylogenetic tree – in biological terms, only the extant species (and not the ancestral/ extinct ones).

Re. Are the 27 language pairs representative of the major subgroups (more than ten of them) of Austronesian? Adding this information would be really useful for the reader.

We have added in the main text a list of the 10 Austronesian subgroups represented in our analyses (page 6). For more information, please see Gray et al. 2009 for a classification of all the languages in the full tree (including all those in our sample) in subfamilies.

Re. Monte Carlo sampling method  Monte Carlo Markov Chain sampling method

The description of Bayesian analyses in the first paragraph of the “Statistical Analyses” is a general overview on how these methods work. Therefore, in that paragraph we simply state that multiple MC sampling methods can be used. Later on (3 paragraphs below), when describing the specific approach we used in this paper, we specify (here now we have added the “Markov”) that we are using three Hamiltonian Monte Carlo Markov chains in parallel.

Re. Poisson link function  Is it a Poisson regression? (generalized linear model). Then, the MCMC sampling is just another way to do the estimation which might be useful in the case of small data.

Exactly.

Re. R^ estimates  What is the statistic that is being referred to?

R^ estimates (or convergence diagnostic) are a way of assessing the convergence of Markov Chains. It compares the between- and within-chain estimates for model parameters and other univariate quantities of interest. If chains have not mixed well (i.e. the between- and within-chain estimates don't agree), R^ is larger than 1. See McElreath et al. 2015.

Re. How are the word losses and gains estimated? is it from an ancestral construction analysis?

One of the main advantages of our approach is that by identifying changes in homologous word sets (see description of vocabulary data) we can estimate separate rates of word gain and loss, whilst using independently established relationships between languages to correct for phylogenetic nonindependence. In other words, we are not using the lexical data for building the phylogeny. Our aim was not to establish the relationships between languages or their general levels of similarity. Instead, we considered the presence or absence of cognates on a pair-wise basis to localize the gain or loss of particular words to the history of each language.

What we do is take two languages in a sister pair, A and B, and count the relative number of word gains and losses that have occurred in each language since they shared a common ancestral language. We consider that, if a lexeme present in either of these languages has a cognate in at least one other member of the language family, then it must have been retained from the ancestral language. If a cognate is present in language A but not in B, then we assume it has been retained in A but lost in B.

Re. How is lexical turnover being computed? Sum of gain and loss or is it estimate from the data?

The lexical turnover was computed as the sum of the gains and losses for each language, yes.

Re. Can the regression be run for more languages only with two factors: geographical isolation and population size? The small number of languages could make a big difference in the interpretations. Also, the languages' sample coverage might contribute to the contradictory results with Greenhill et al. If it is a larger dataset, then, the regression without the conflict variables might support what Greenhill et al. found.

Since our trimmed phylogenetic tree only comprises variables included in Pulotu (our estimate of pre-contact population size is that in Pulotu as well) – our maximum number of languages for which we have gain and loss data, as well as phylogenetic branch lengths to extract sister pairs are this sample. We welcome future studies that consider other variables, in other language families or in larger databases and that contribute to getting closer to understanding how it is that the worlds’ linguistic diversity is so vast and dynamic. However, one of the main aims in this study was to try to disentangle the effect of population size and population structure on linguistic evolution as both these variables, together with isolation contribute to the effective population size that exchanges linguistic information. This aim was further justified after our model-selection criteria indicated that the best model was that including both population size and conflict. That is, our aim was not to replicate Greenhill et al.’s study. We believe the methods they used were robust, yet we attempt to provide an explanation for their findings and shed light on a new dimension of factors that have been widely hypothesised by linguists to contribute to linguistic evolution but that have been largely ignored from quantitative studies.

We thank the reviewer again for the comments and hope the replies have clarified the issues.

#####

Once again, we would like to thank the editor for organising the review process and we do hope you are satisfied with the changes we made according to your recommendations.

Please let us know if there is anything else you would like us to revise. We look forward to your decision.

Best,

Cecilia Padilla-Iglesias, Erik Gjesfjeld and Lucio Vinicius

---

## [Decision Letter · Decision Letter 1]

11 Nov 2020

PONE-D-20-18721R1

Geographical and social isolation drive the evolution of Austronesian languages

PLOS ONE

Dear Dr. Padilla-Iglesias,

Thank you for submitting your manuscript to PLOS ONE. After careful consideration, we feel that it has merit but does not fully meet PLOS ONE’s publication criteria as it currently stands. Therefore, we invite you to submit a revised version of the manuscript that addresses the points raised during the review process.

Please have a look at the revisions and clarifications proposed by reviewer #2

We look forward to receiving your revised manuscript.

Kind regards,

Francesc Calafell

Academic Editor

PLOS ONE

Reviewers' comments:

Reviewer's Responses to Questions

**Comments to the Author**

1. If the authors have adequately addressed your comments raised in a previous round of review and you feel that this manuscript is now acceptable for publication, you may indicate that here to bypass the “Comments to the Author” section, enter your conflict of interest statement in the “Confidential to Editor” section, and submit your "Accept" recommendation.

Reviewer #1: All comments have been addressed

Reviewer #2: (No Response)

Reviewer #3: All comments have been addressed

2. Is the manuscript technically sound, and do the data support the conclusions?

Reviewer #1: Yes

Reviewer #2: Yes

Reviewer #3: Yes

3. Has the statistical analysis been performed appropriately and rigorously? 

Reviewer #1: Yes

Reviewer #2: Yes

Reviewer #3: Yes

4. Have the authors made all data underlying the findings in their manuscript fully available?

Reviewer #1: Yes

Reviewer #2: Yes

Reviewer #3: Yes

5. Is the manuscript presented in an intelligible fashion and written in standard English?

Reviewer #1: Yes

Reviewer #2: Yes

Reviewer #3: Yes

6. Review Comments to the Author

Reviewer #1: I am satisfied with the corrections and recommend publication. I have no additional comments for the authors at this time.

Reviewer #2: The manuscript had improved notably from the first round. I, however, still had some, mainly minor comments on how to improve the manuscript. There are also some a bit more major points addressed concerning the discussion section.

Introduction

Introduction, in the end of the first paragraph you write: “Nonetheless, given a vast corpus of research has emerged on the evolutionary, ecological and social correlates of the global distribution of linguistic diversity.” Should the word “given” be deleted from the sentence? It would make more sense to me that way.

To me it sounds that in the Thomason example the point in changing the language is not so much in cultural group marking but instead in making the language mutually unintelligible for warfare reasons: “-- so that the enemies wouldn’t understand them”. Would the intentional changes in the language of Delaware Indians apply only in situations where they are in contact with Iroquois? Please clarify whether it is more about mutual unintelligibility or cultural group marking.

Arelated  a related

It would be clearer to present the study area “Here we provide the first integrated test of the effect of various sociodemographic and geographic features on linguistic diversification among 50 Austronesian languages.” before mentioning the problems of in using contemporary speaker populations as the data. In the current version the study area is first mentioned as an example case where using contemporary populations as the data is problematic.

Materials and methods

In table S1 there are 27 sister pairs listed even though based on the text there should be only 25 of them. In addition, it would be good to add to the Table S1 or to the Figure 1 which pairs belong to which of the 10 major subgroups of the Austronesian family (listed in the text) as this is not clear to a reader who is not familiar with the family. Please fix these.

“(ii) using only tip branches also avoids the problem of non-independence between ancestor and descendant lineages within the phylogeny, as each branch is likely to be more similar in many traits to its immediate neighbours than to more distantly related branches”. It is not necessarily clear to the reader how similarity of immediate neighbours avoids the problem of non-independence. This was clearly explained earlier in the text but it would be good if you could clarify also this bit here.

In section Social, demographic and geographic variables you write “Data on population size, geographical isolation and conflict within and between cultures--“ It first seems that the list has only four variables instead of the five included in the study and the reader may think whether one of the variables were taken from some other source. However, if all of them were from this source but the two between-cultures variables were grouped together, it would be good to clarify which is the case.

“This selection process resulted in 27 pairs (n=54 languages) of Austronesian languages.” To make sure the reader is not confused with the number of language pairs used in this study, it would be good to note here again why the final number of pairs was 25 with reference “see above”.

Table S6 was not included in the Supplementary Material. Please add this.

“If a word form found in one sister language has a cognate in other languages in the language family, it is likely to have been inherited from the common ancestor. This implies that the absence of that cognate form in the other sister language must be due to its loss after divergence from their exclusive common ancestor.” Would you please clarify is the whole phylogeny or only the sub-branch in question taken into account when checking the commonness of a certain cognate when counting the rates of word gains and losses? It sounds like a quite a rough requirement if the cognate form needs to be found from all the other languages of the family except from one of the two sister branches. Could you elaborate this?

“We did not include any identified loan words in the analysis--“ Does this mean that some of the 210 basic semantic units were excluded from some sister pair comparisons because there were borrowings? What was the source of information about the existence of borrowings? It would be good to clarify this so the reader knows e.g. how large part of the data was excluded due to the existence of loanwords.

“The total number of gains, losses, and non-informative results--“ Please elaborate what are these “non-informative results”. Please also add to this section how the lexical turnover was calculated.

Parethesis is missing after the list of five predictors in the second paragraph of Statistical analysis section.

Discussion

You say in the introduction that “Here we provide the first integrated test of the effect of various sociodemographic and geographic features on linguistic diversification--“. However, as none of these variables acts in isolation (e.g. some isolated groups are small some large, some small isolated groups have a lot of conflict with other cultures while some may have only little), it would be very relevant to study and discuss also the interactions between the studied variables. For example, you write “We found that being geographically isolated (--) greatly speeded up the rate of word losses and also moderately that of word gains. This is consistent with the idea that geographical isolation results in a higher risk of random loss of cultural items due to incomplete inter-generational sampling of existing variation” Does this inference of incomplete inter-generational sampling apply for both small and large populations? This is important especially as in the discussion in the population size paragraph you write “--smaller populations should be more prone word losses due to random sampling effects [16,17,19,23]--”

“In summary we argue that contact has contributed to differentiation among Austronesian languages mostly by preventing the random loss of existing lexical items, despite also hindering the ability of novel lexical innovations to emerge and be utilised”. I don’t quite see why you argue that contact has contributed to the differentiation if you say that contact both prevents the random loss of lexical items and hinders novel lexical innovations. To me this sounds that when contact takes place, differentiation does not happen. Please clarify this.

When using only historical population size as predictor, your results for word gains were: estimate= 0.09, 90% HPDI: [0.01, 0.18] and for word losses: estimate = 0.01, 90% HPDI: [-0.10, 0.12]. When using the contemporary population size the results obtained with your method were for word gains: 0.06, 90% HPDI: [-0.06, 0.17] and for word losses: = 0.01, 90% HPDI: [-0.10, 0.12]. The result for word loss is exactly the same suggesting that the usage of historical vs. contemporary population size data does explain the difference between your and Greenhill et al’s results concerning word losses. In addition, the difference in the results of word gains also looks very minimal even though it changes a significant result obtained with the historical population size to non-significant when using contemporary population size. These issues should be taken into account when discussing the possible reasons why your results differ from those of Greenhill et al’s.

Related to the previous comment, it would be good to mention how much smaller your data were compared to Greenhill et al’s.

I believe reference to Fig 2 in the end of the fifth paragraph of Discussion should be Fig 1.

Reviewer #3: The authors made great effort to address the comments. I am completely satisfied with the responses and the revised manuscript.

7. PLOS authors have the option to publish the peer review history of their article (what does this mean?). If published, this will include your full peer review and any attached files.

Reviewer #1: No

Reviewer #2: **Yes: **Terhi Honkola

Reviewer #3: No

---

## [Author Response · Author response to Decision Letter 1]

14 Nov 2020

Dear Dr Calafell

Thank you very much for giving us the opportunity to further revise our manuscript. We now hope to have addressed all the additional questions raised by Reviewer 2 and hope that you are satisfied to with the result. 

Below we specify how we have modified the Text and Supplementary Materials according to the specific comments by the reviewer.

Re. In the end of the first paragraph you write: “Nonetheless, given a vast corpus of research has emerged on the evolutionary, ecological and social correlates of the global distribution of linguistic diversity.” Should the word “given” be deleted from the sentence? It would make more sense to me that way.

Thank you very much for spotting this. It has now been corrected.

Re. To me it sounds that in the Thomason example the point in changing the language is not so much in cultural group marking but instead in making the language mutually unintelligible for warfare reasons: “-- so that the enemies wouldn’t understand them”. Would the intentional changes in the language of Delaware Indians apply only in situations where they are in contact with Iroquois? Please clarify whether it is more about mutual unintelligibility or cultural group marking.

There are a whole suit of examples along the same lines in the cited reference of Thomason (2007) where she explicitly relates this “cultural group marking” use of language with deliberate changes in grammar and phonology. Warfare is an extreme example of the phenomenon, where cultural groups with similar languages (otherwise changes would not be necessary) modify them so as to preserve inter-group but reduce inter-group intelligibility. The idea here is just like in the Nettle and Dunbar 1997 model – in situations of conflict, speakers of opposing sides (particularly when speaking related languages) will deliberately plan to modify their language fast enough so that enemies won’t be able to speak or understand their language, and by doing so, be able to pretend to be allies. The result is of course more differentiated languages, and thus stronger cultural markers.

Re. Arelated  a related

This has been amended.

Re. It would be clearer to present the study area “Here we provide the first integrated test of the effect of various sociodemographic and geographic features on linguistic diversification among 50 Austronesian languages.” before mentioning the problems of in using contemporary speaker populations as the data. In the current version the study area is first mentioned as an example case where using contemporary populations as the data is problematic.

We thank the reviewer for the suggestion. However, we believe it makes more sense to introduce the question we are addressing first, and then our approach and dataset. This is why both in the abstract and main text we first make an effort to show that the puzzle of linguistic differentiation that the factors driving it remain unsolved. This is in fact the overall justification or our study; the choice of data is a subsequent step, which as mentioned by the reviewer, was affected by issues regarding data/population suitability. 

We tried to think of a way of introducing the information about our case study earlier, but we believe that this would break our line of argumentation. However, we mention Austronesian languages in the title and on line 5 of the abstract, which we believe makes our case study and area clear to readers from the very beginning. 

Re. In table S1 there are 27 sister pairs listed even though based on the text there should be only 25 of them. In addition, it would be good to add to the Table S1 or to the Figure 1 which pairs belong to which of the 10 major subgroups of the Austronesian family (listed in the text) as this is not clear to a reader who is not familiar with the family. Please fix these.

The two pairs corresponding to the branch lengths that represent the whole tree have now been removed (and therefore Table S1 only contains 25 pairs). We have also added an extra column specifying the Austronesian subgroups (according to those in the tree by Gray et al. 2009) to which each of the languages belong to.

Thanks for spotting it.

Re. “(ii) using only tip branches also avoids the problem of non-independence between ancestor and descendant lineages within the phylogeny, as each branch is likely to be more similar in many traits to its immediate neighbours than to more distantly related branches”. It is not necessarily clear to the reader how similarity of immediate neighbours avoids the problem of non-independence. This was clearly explained earlier in the text but it would be good if you could clarify also this bit here.

In other words, the problem of non-independence due to relatedness of the branches. We have now added this to the paragraph, which now reads as:

The sister-pairs approach has two main advantages over whole tree phylogenetic methods that use every branch in a phylogeny as a datapoint in an analysis, namely: (i) using only the tips of the phylogeny avoids the need to infer less reliable ancestral states down the phylogeny, which is particularly important given that some Austronesian languages date back as far as 4,500 years ago; and (ii) using only tip branches also avoids the problem of non-independence between ancestor and descendant lineages within the phylogeny, as each branch is likely to be more similar in many traits to its immediate neighbours than to more distantly related branches simply due to relatedness [22].

Re. In section Social, demographic and geographic variables you write “Data on population size, geographical isolation and conflict within and between cultures--“ It first seems that the list has only four variables instead of the five included in the study and the reader may think whether one of the variables were taken from some other source. However, if all of them were from this source but the two between-cultures variables were grouped together, it would be good to clarify which is the case.

No variables were grouped together, and indeed all of them were taken from the same source. We have amended that sentence and hope to have made that clearer. It now reads as 

All social, geographic and demographic data used for our analyses was obtained from the “traditional state” section in Pulotu database, which concerns the state of cultural groups prior to large-scale modernisation [82].

Re. “This selection process resulted in 27 pairs (n=54 languages) of Austronesian languages.” To make sure the reader is not confused with the number of language pairs used in this study, it would be good to note here again why the final number of pairs was 25 with reference “see above”.

This has now been done.

Re. Table S6 was not included in the Supplementary Material. Please add this.

Sorry about that. This has now been included.

Re. “If a word form found in one sister language has a cognate in other languages in the language family, it is likely to have been inherited from the common ancestor. This implies that the absence of that cognate form in the other sister language must be due to its loss after divergence from their exclusive common ancestor.” Would you please clarify is the whole phylogeny or only the sub-branch in question taken into account when checking the commonness of a certain cognate when counting the rates of word gains and losses? It sounds like a quite a rough requirement if the cognate form needs to be found from all the other languages of the family except from one of the two sister branches. Could you elaborate this?

Thanks. We clarify again that the classification of cognates (in other words, the 210 basic semantic units) was not done for the present study but were taken from a peer-reviewed, established database (the Austronesian Basic Vocabulary Database): https://abvd.shh.mpg.de/austronesian/, detailed in Greenhill et al. (2008). The authors of the database determined which words represented cognates across the whole phylogeny, as well as which words represented borrowings in each of the languages (and thus needed to be excluded from the cognate lists of each of the languages).

Re. “We did not include any identified loan words in the analysis--“ Does this mean that some of the 210 basic semantic units were excluded from some sister pair comparisons because there were borrowings? What was the source of information about the existence of borrowings? It would be good to clarify this so the reader knows e.g. how large part of the data was excluded due to the existence of loanwords.

Please see the comment above.

Re. “The total number of gains, losses, and non-informative results--“ Please elaborate what are these “non-informative results”. Please also add to this section how the lexical turnover was calculated.

This has been modified accordingly, and we have moved the description of the lexical turnover variable from the statistical analysis section to this paragraph. Now it reads as

The total number of gains and losses were counted for all available semantic categories for each pair of languages using the Python package RateCounter developed by Simon Greenhill (https://github.com/SimonGreenhill/RateCounter; see [22] and Table S5). The overall rate of lexical turnover was then computed for each language pair by adding the number of gains and losses.

Re. Parethesis is missing after the list of five predictors in the second paragraph of Statistical analysis section.

Thank you for your attention, we have added the parenthesis.

Discussion

Re. You say in the introduction that “Here we provide the first integrated test of the effect of various sociodemographic and geographic features on linguistic diversification--“. However, as none of these variables acts in isolation (e.g. some isolated groups are small some large, some small isolated groups have a lot of conflict with other cultures while some may have only little), it would be very relevant to study and discuss also the interactions between the studied variables. For example, you write “We found that being geographically isolated (--) greatly speeded up the rate of word losses and also moderately that of word gains. This is consistent with the idea that geographical isolation results in a higher risk of random loss of cultural items due to incomplete inter-generational sampling of existing variation” Does this inference of incomplete inter-generational sampling apply for both small and large populations? This is important especially as in the discussion in the population size paragraph you write “--smaller populations should be more prone word losses due to random sampling effects [16,17,19,23]--”

This is a relevant question. The fact that our results show that both population size and geographical isolation exert a positive effect on the rate of word loss alongside the fact that those two variables are not correlated, indeed indicate that ceteris paribus, more isolated languages will lose terms faster (regardless of their speaker population size) and that languages with less speakers (regardless of how geographically isolated they are) will lose terms faster. However, whether the two variables together generate a positive feedback (i.e. interact) affecting the magnitude of the speed at which this happens is a question that would require a different analysis, as with the sister-pairs approach we cannot estimate rates of change of single languages but relative rates of change of sister-pairs compared to each other. 

Although we believe that potentially all factors could interact with one another, such analysis goes beyond the scope of the present study due to a lack of both methodology and data. Here, we wanted to test the simplest possible model, that is, whether each of those variables exerted independent, additive effects on rates of lexical change. We thank the reviewer for the suggestion, which we may address in future studies.

Re. “In summary we argue that contact has contributed to differentiation among Austronesian languages mostly by preventing the random loss of existing lexical items, despite also hindering the ability of novel lexical innovations to emerge and be utilised”. I don’t quite see why you argue that contact has contributed to the differentiation if you say that contact both prevents the random loss of lexical items and hinders novel lexical innovations. To me this sounds that when contact takes place, differentiation does not happen. Please clarify this.

Thanks for pointing this out. This was a mistake. We wanted to say geographical isolation (or lack of contact) instead of contact here. We replaced it in the text and the sentence makes sense now:

In summary, we argue that geographical isolation may have contributed to differentiation among Austronesian languages mostly by preventing the random loss of existing lexical items, despite also hindering the ability of novel lexical innovations to emerge and be utilised [38,59], rather than by affecting the internal social dynamics of isolated groups

Re. When using only historical population size as predictor, your results for word gains were: estimate= 0.09, 90% HPDI: [0.01, 0.18] and for word losses: estimate = 0.01, 90% HPDI: [-0.10, 0.12]. When using the contemporary population size the results obtained with your method were for word gains: 0.06, 90% HPDI: [-0.06, 0.17] and for word losses: = 0.01, 90% HPDI: [-0.10, 0.12]. The result for word loss is exactly the same suggesting that the usage of historical vs. contemporary population size data does explain the difference between your and Greenhill et al’s results concerning word losses. In addition, the difference in the results of word gains also looks very minimal even though it changes a significant result obtained with the historical population size to non-significant when using contemporary population size. These issues should be taken into account when discussing the possible reasons why your results differ from those of Greenhill et al’s.

Again, sorry about that and thank you for pointing it out. We had made a typo which has now been corrected. However, in effect discrepancies cannot be attributed to choice of statistical methods. We think that it is a combination of a) only including population size and b) the choice of the variable.

Re. Related to the previous comment, it would be good to mention how much smaller your data were compared to Greenhill et al’s.

The studies by Bromham et al. 2015 and Greenhill et al. 2010 included 10 and 81 language pairs respectively. We have now added this to the main text.

Re. I believe reference to Fig 2 in the end of the fifth paragraph of Discussion should be Fig 1.

Indeed. Thank you for pointing that out.

We would like to thank you very much again and for the very detailed and useful comments on our manuscript.

Best,

Cecilia Padilla-Iglesias, Erik Gjesfjeld and Lucio Vinicius

---

## [Editor Report · Decision Letter 2]

17 Nov 2020

Geographical and social isolation drive the evolution of Austronesian languages

PONE-D-20-18721R2

Dear Dr. Padilla-Iglesias,

We’re pleased to inform you that your manuscript has been judged scientifically suitable for publication and will be formally accepted for publication once it meets all outstanding technical requirements.

Kind regards,

Francesc Calafell

Academic Editor

PLOS ONE
---

## [Editor Report · Acceptance letter]

19 Nov 2020

PONE-D-20-18721R2 

Geographical and social isolation drive the evolution of Austronesian languages 

Dear Dr. Padilla-Iglesias:

I'm pleased to inform you that your manuscript has been deemed suitable for publication in PLOS ONE. Congratulations! Your manuscript is now with our production department. 

Kind regards, 

on behalf of

Dr. Francesc Calafell 

Academic Editor

PLOS ONE